# Isolation and Characterization of Equine Lymph Node Endothelial Cells

**DOI:** 10.3390/vetsci12090905

**Published:** 2025-09-18

**Authors:** Tomas Lugo, Stephanie Myers, Thu Annelise Nguyen

**Affiliations:** School of Veterinary Medicine, Texas Tech University, Amarillo, TX 79106, USA; tomas.lugo@ttu.edu (T.L.); annelise.nguyen@ttu.edu (T.A.N.)

**Keywords:** multisystemic eosinophilic epitheliotropic disease, lymph nodes, primary cells, endothelial cells, equine

## Abstract

This study focuses on the characterization of an isolated population of cells from a 3-year-old horse that was diagnosed with multisystemic eosinophilic epitheliotropic disease (MEED). MEED is a rare, long-term illness that leads to weight loss, hair loss, and skin ulcers. There is very little known about what causes MEED, and current research is limited. The results showed that we successfully established a primary cell culture and subsequently identified significant changes via genomic profiling. This provides the field of equine health with a model for understanding the complexity of immune diseases.

## 1. Introduction

Lymph nodes are complex structures involved in the immune system of most mammalian species [1,2,3]. They are made up of parenchyma containing lymphocytes and other immune cells, such as natural killer cells and macrophages [4]. This cellular component is supported by a stromal framework that provides structural integrity. The parenchyma is organized into distinct regions which are the cortex, containing lymphoid follicles; the paracortex; and the medulla [5]. Lymph nodes function as filters for lymphatic fluid, which allows for monitoring pathogens, and serve as sites for the activation and proliferation of immune cells [1,4]. These structures are central to immunological functions, and their disruption can play a role in or give insight into rare immune diseases such as multisystemic eosinophilic epitheliotropic disease (MEED), where abnormal immune cell activity contributes to widespread tissue infiltration [4,6].

Understanding the lymph node architecture and function provides a critical foundation for laboratory-based approaches to study immune responses [7]. In vitro studies are vital in studying molecular mechanisms and biological pathways based on the ability to have direct control of experimental variables. In vitro mammalian model systems utilize primary cells, cell lines, cancer cells, or a mixture of these cells to conduct experiments [7,8,9,10,11]. Primary cells most closely resemble their tissue of origin but have limited lifespans, while immortalized and cancer-derived lines can proliferate indefinitely but often differ biologically from normal cells [9,12].

In equine health research, there is a striking lack of relevant cell models, particularly for immune-associated cell types. The American Type Culture Collection (ATCC) currently lists only one equine cell line, NBL-6, a fibroblast line whose properties differ markedly from those of lymphoid endothelial cells. While multiple laboratories have attempted to culture primary equine cells, no commercially available lymph node-derived cell lines currently exist [13,14,15,16]. This gap limits mechanistic studies of equine immune disorders, especially those involving lymphoid tissue pathology.

Therefore, in this study, we have established a primary cell culture from an equine lymph node obtained from a horse diagnosed with MEED. The tissue was collected from a 3-year-old horse diagnosed with MEED, a rare, immune-mediated disease characterized by excessive eosinophil production and infiltration into multiple organs, including lymphoid tissue, intestines, and liver [17,18,19,20,21]. MEED presents with diverse clinical signs such as exudative dermatitis, alopecia, skin lesions, chronic diarrhea, and other systemic manifestations, and at this time, remains poorly understood, with only a few cases reports available in the literature [17,19,20,21,22]. In addition to these clinical signs, reports in certain horse breeds have documented lesions in the intestines, stomach, and cecum. Histopathological examination of surgically excised tissue revealed lymphoplasmacytic infiltration, and immunohistochemistry demonstrated CD3-positive cell populations [17,19,21]. In contrast, three reported cases in donkeys did not describe gastric lesions; instead, lesions were observed in the colon and colonic lymph nodes [21]. Overall, MEED pathogenesis remains unknown, and current treatment options have shown limited long-term success.

Establishing an endothelial primary cell culture, derived from lymph nodes of horses afflicted by MEED, provides a disease-relevant platform for equine immunology research. MEED involves eosinophil infiltration and inflammation within lymphoid tissues, meaning the source of these cells is directly pertinent to immune endothelial interactions and disease mechanisms. In this study, we describe the isolation, culture, and characterization of this population of cells, including growth curve and media optimization, biomarker expression profiling, and genetic sequencing. The aim of this study was to establish the first physiologically relevant in vitro system of equine lymph nodes, and it offers a novel platform for understanding MEED pathogenesis.

## 2. Methodology

### 2.1. Primary Cell Extraction

Tissue for the primary cell culture was extracted from a horse diagnosed with multisystemic eosinophilic epitheliotropic disease. A board-certified pathologist at the Texas Tech School of Veterinary Department extracted lymphatic tissue from various regions of the body.

Lymph node tissues were taken from each of the following regions: submandibular, mesenteric, lumbar, left inguinal, and right inguinal. The tissue samples were immediately transferred from the pathology department to the cell culture laboratory in five separate Eppendorf tubes (Nunc™ 15 mL and 50 mL Conical Sterile Polypropylene Centrifuge Tubes, Thermo Fisher Scientific, Inc., Waltham, MA, USA), each containing 10 mL of 1× phosphate-buffered saline (PBS) on ice.

The lymph node tissues were washed four times with PBS at room temperature, completely submerging the tissue using Adson Tissue Forceps in a biosafety hood. The lymph node tissues were then separated from connective and fat tissues using sterilized Surgimac 4.75′′ Castroviejo Surgical Scissors (Securos Surgical, Tuttlingen, Germany) and a steel surgical scalpel (Securos Surgical, Tuttlingen, Germany) with a size 10 blade. The tissues were minced into 0.5 mm × 0.5 mm pieces using the same tools. The remaining tissue was then submerged in 20 mL of either Trypsin without phenol red (1×), Collagenase Type IV, or a 1:1 combination of both in a 50 mL Eppendorf tube (TrypLE™ Express Enzyme (1×), no phenol red, Gibco; Thermo Fisher Scientific, Inc., Waltham, MA, USA; Collagenase Type IV, Sigma-Aldrich, Darmstadt, Germany).

The tissue was then incubated in a water bath at 37 °C for 20 min. Following incubation, the Eppendorf tube was returned to the biosafety cabinet, and the contents were filtered using a 40 μm Cole-Parmer Essentials Cell Strainer (Cole-Parmer Essentials Cell Strainer, Cole-Parmer, East Bunker Ct Vernon Hills, IL, USA). The filtered contents were then centrifuged at 5000 rpm for 5 min. The supernatant was discarded without disturbing the pellet. The pellet was re-suspended in Roswell Park Memorial Institute Medium 1640 (RPMI 1640 Medium; Gibco; Thermo Fisher Scientific, Inc., Waltham, MA, USA) containing 10% fetal bovine serum (FBS; Invitrogen; Thermo Fisher Scientific, Inc., Waltham, MA, USA) and seeded into a culture-treated T-25 flask. To prevent bacterial and fungal growth, 1% antibiotic–antimycotic was added to the medium (stock Antibiotic-Antimycotic at 100× concentration; Gibco; Thermo Fisher Scientific, Inc., Waltham, MA, USA).

### 2.2. Cell Monitoring and Contamination Control

Imaging and observation were performed and notes made on morphological changes over time and any signs of contamination. Medium was replaced every 2 days, and cells were washed twice weekly with PBS. After two weeks, stable cell growth was observed. Before subculturing or passaging, the cells were left to grow to 80–90% confluency prior to subculturing. The flask was kept in a 5% CO_2_ incubator at 37 °C. After reaching 90% confluency, cells were frozen and kept in a −150 °C freezer.

### 2.3. Trypsin Differentiation

Cells that were in passage 3 were differentiated using trypsin express enzyme 1× for 1 min in an incubator at 37 °C with 5% CO_2_. The supernatant was discarded, and the flask was washed with PBS. Cells were then re-submerged with the corresponding media.

### 2.4. Growth Curve Assay

A growth curve assay was carried out to evaluate the cell proliferation rate and cell viability in 7 days. Cells were seeded in a 24-well plate that was treated for a cell culture at a starting point of 500,000 cells per well. The starting volume used was 800 µL with RPMI 1640 media. Cells that were used in this growth curve assay were in passage 5. Cells were counted and observed using an inverted microscope at 4× magnification. Cell viability was assessed using a trypan blue exclusion assay (Nexcelom Auto2000 cell counter, Lawrence, Inglewood, CA, USA).

### 2.5. Growth Media Curve Assay

The growth curve assay was conducted with 4 types of media: Dulbecco’s modified Eagle’s medium F-12 (DMEM F-12; Gibco; Thermo Fisher Scientific, Inc., Waltham, MA, USA), Roswell Park Memorial Institute Medium 1640 (RPMI 1640 Medium; Gibco; Thermo Fisher Scientific, Inc., Waltham, MA, USA), Dulbecco’s Modified Eagle Medium (DMEM; Gibco; Thermo Fisher Scientific, Inc., Waltham, MA, USA), and Minimum Essential Medium (MEM; Gibco; Thermo Fisher Scientific, Inc., Waltham, MA, USA). Cell viability was assessed using trypan blue exclusion assay (Nexcelom Auto2000 cell counter).

### 2.6. Cell Characterization Using Immunofluorescence Assay

Cells were plated in a 24-well plate with a coated coverslip (Nunc™ Thermanox™ Coverslips; Thermo Fisher Scientific, Inc., Waltham, MA, USA). These cells were incubated using DMEM F-12 at 37 °C and 5% CO_2_ in the incubator until reaching 70% confluency. Once reaching 70% confluency, cells were washed with PBS for 3 times and subsequently fixed in 4% paraformaldehyde in PBS for 20 min at room temperature. The fixed cells were then washed 3 times with PBS and were permeabilized with 0.25% of Triton X-100 in PBS for 20 min at room temperature. The permeabilized cells were then washed 3 times with PBS and blocked with 3% BSA in PBS for 2 h at room temperature. An immunofluorescence assay was performed using primary antibodies against CD3, E-cadherin, CD31, LYE1, Vimentin, ZO1, Beta-Catherin, CD79, IBA1, and LY6G at 1:500 dilution. After 2 h in blocking buffer, cells with primary antibodies in blocking buffer were incubated overnight. Cells were washed with blocking buffer 3 times and incubated with Alexa fluor-conjugated secondary antibodies for 2 h at room temperature. Again, cells were washed 3 times with blocking buffer and incubated with DAPI at 1:1000 dilution for 10 min. After incubation, cells were washed once with blocking buffer. The 24-well plate with the coverslip was then mounted to the microscope slide by removing the coverslip and mounting the coverslip with a drop of ProLong™ Gold Antifade Mountant (Invitrogen; Thermo Fisher Scientific, Inc., Waltham, MA, USA).

### 2.7. Cell Contamination Assay

Cells were tested for contamination in passage 3 using a cell culture contamination detection kit from Invitrogen (Cat. No. C7028, Invitrogen, Waltham, MA, USA). This kit tested for any Gram-positive or Gram-negative bacteria, and tested for yeast contamination using fluorescent dye. In addition, mycoplasma contamination was assessed using a commercial kit from Merdian Bioscience (Cat. No. 14-910-108, Meridian Bioscience, Cincinnati, OH, USA).

### 2.8. Whole-Genome Sequencing

#### 2.8.1. DNA Isolation

Primary cells were cultured using the RPMI, and we used passage 5 to isolate DNA using QIAamp DNA Micro Kit (Cat. No. 56304, QIAGEN, Hilden, Germany). The quality and quantity of DNA were analyzed using an *Invitrogen™ Nanodrop™ One Spectrophotometer* (Thermo Fisher Scientific, Waltham, MA, USA) and *Invitrogen™ Qubit™ 4 Fluorometer* (Thermo Fisher Scientific, Waltham, MA, USA).

#### 2.8.2. Library Preparation and Sequencing

Genomic DNA was used to make a 150-base-pair paired-end library (TruSeq DNA PCR-Free kits, Illumina, San Diego, CA, USA) and sequenced on a NextSeq500, with a quality score of greater than 20 achieved. The raw reads were filtered to remove low-quality reads and adaptor sequences. These filtered reads were then analyzed using the fastqc program [17]. After analyzing the quality of the reads, the reference genome was indexed using bowtie2 [18] and subsequently mapped to the NIH reference genome, EquCab3.0, using Bowtie2. After alignment to the reference genome, the resulting output was converted to BAM format using the Samtools function. The BAM files were then sorted using Samtools. Variant detection was performed using the Next Generation Sequencing Eclipse Plugin (NGSEP) [19]. Initial variant calling was conducted using the EquCab3.0 reference genome and high-quality BAM files, followed by variant annotation using a genomic feature file (GFF). Several filtering steps were applied to enhance data quality, including filtering based on variant quality scores and read depth thresholds. Summary statistics were then generated to provide an overview of the identified and annotated variants.

## 3. Results

Of the five lymph node tissues that were processed, only one provided a sufficient number of cells and was subsequently used to perform cell expansion and trypsinization, leading to the successful culturing of lymph node cells, originating from the horse’s mesenteric region.

### 3.1. Cell Culture

These primary cells generally reach 80% confluency within 9 to 12 days of culturing. Cell culturing revealed a mixture of fibroblastic and epithelial-shaped cells. In the initial days after seeding, the cells displayed a spindle-shaped morphology (Figure 1). However, after 5 to 7 days, the cells adopted an epithelioid shape, as shown in Figure 2B. Upon reaching 80% to 90% confluency, there was a mixture of spindle-shaped and epithelioid-shaped cells (Figure 1). There were slight differences in morphology between different passage numbers. Passage 5 exhibited a high number of epithelioid-shaped cells (Figure 2B). However, in passage 7 and passage 8, there were fewer epithelioid cells corresponding to the passage number, as shown in Figure 2.

### 3.2. Growth Curve Assay Results

The results of the growth curve assays are presented in Table 1 and Figure 3 and Figure 4. Starting with a seed density of 500,000 cells per well, observations were made over 7 days using an inverted microscope. By the end of this period, the highest cell count registered was 4.25 million cells with an average viability of 77%. This represents an average cellular proliferation rate of 22,000 cells per hour over the week-long experiment. This experiment provides insight into the proliferation dynamics of these primary cells over a seven-day timeframe.

### 3.3. Growth Curve Media Assay

The optimal growth medium was assessed using four distinct medium types. Among them, RPMI consistently exhibited the most stable and uniform growth compared to the others. RPMI maintained a steady cellular growth path throughout the seven-day experiment, showing no decline after the fourth day. Conversely, MEM displayed significant fluctuations and little to no cellular growth after the fourth day (Figure 5). DMEM and DMEM-F-12 yielded similar outcomes, with growth plateauing after the sixth day. DMEM exhibited an increase in cell mortality compared to DMEM F-12. By end of the experiment, RPMI was the preferred medium in terms of cell viability and cell growth, registering a 76% viability on the final day (Figure 5). The datasets have been visualized in Figure 5, Figure 6, Figure 7 and Figure 8, with details given in Table 2.

### 3.4. Cell Characterization Using Immunofluorescence Assay Results

Immunofluorescence staining revealed distinct expression patterns for the selected biomarkers in the primary cell culture derived from lymph nodes (Table 3, Figure 9, Figure 10 and Figure 11). Strong positive fluorescence signals were observed for the endothelial-associated markers CD31 and ZO-1, indicating vascular endothelial identity and a tight junction presence. E-cadherin and β-catenin staining produced clear membrane-associated fluorescence, consistent with adherens junction structures. Vimentin staining yielded intense cytoplasmic signals, consistent with mesenchymal cytoskeletal components. LYVE1 staining was conducted in a subset of cells, and CD79 also produced positive fluorescence. In contrast, no detectable fluorescence was observed for CD3, IBA1, or LY6G.

### 3.5. Whole-Genome Sequencing Results

Quality control of the raw sequencing data was performed using FastQC version 0.12.0. A total of 2.5 Gb of sequence data was generated, with an average read length of 151 bp. The GC content was approximately 43%, which is within the expected range for the species. The per-base sequence quality remained high across all reads, with the majority of bases scoring above Q30, indicating an excellent read accuracy (Figure 12 and Figure 13). The GC content was measured at 43%, consistent with the expected range for the species. The N content was negligible across all positions, with almost no ambiguous base calls (N), highlighting the high accuracy of base identification. The sequence content stabilized after the first few bases, showing a consistent base composition throughout most of the read, with only minor fluctuations at the start and end. Analysis of sequence duplication levels indicated that 81.11% of sequences would remain after deduplication.

A total of 16,855,223 paired reads were processed, with an overall alignment rate of 98.88%. Of these, 22.30% did not align concordantly, while 61.38% aligned concordantly once, and 16.32% aligned concordantly more than once. Among the unaligned pairs, 73.14% aligned discordantly (Table 4 and Table 5). For the remaining unaligned pairs, 18.79% of the individual reads did not align, 16.02% aligned once, and 65.20% aligned more than once.

Variant calling results showed 1,202,674 biallelic SNVs, 264,537 biallelic indels, and 1366 biallelic short tandem repeats (STRs). Following the application of stringent quality and depth filtering criteria, the variant set was significantly reduced to 4859 SNVs, 316 indels, and 23 STRs (Table 5, Appendix A). The depth filtering criteria were set to a depth of 30 and quality score of 20. Key coding variants were preserved, including 107,385 coding SNVs and 23,754 coding indels from the original set (Table 6). These retained variants encompass functionally relevant categories such as synonymous, missense, stop-gained, and frameshift variants. In addition, multiallelic variants were reduced to 25 SNVs, 14 indels, and 10 STRs post-filtering.

## 4. Discussion

In vitro studies play a pivotal role in drug testing, evaluating specific pathways, and modeling diseases [7,10,11,23,24,25,26]. Both cell lines and primary cells provide the foundation of most in vitro research. These models bridge critical knowledge gaps in equine biology; however, there is pronounced shortage of equine primary cell cultures [13,14,15,16,25,27,28]. Due to this scarcity, many researchers resort to immortalized or cancer-derived cell lines, such as e-CAS cells [14,29]. Cancer cells differ markedly from primary cells in physiology, signaling, and environmental responsiveness, while immortalized lines often display altered proliferation rates and unintended mutations, which may affect experimental outcomes [12,30,31]. The misidentification of e-CAS cells originally marketed as equine macrophages but later shown to be derived from mouse macrophages further underscores the challenges of ensuring model accuracy in equine research.

Untransformed primary culturing, derived from lymph nodes, addresses the gap in knowledge by providing a relevant physiological model system. The source of tissue was obtained from a horse diagnosed with MEED; therefore, this model offers unique value for investigating immune-mediated pathology within lymphoid tissue. MEED is characterized by eosinophilic infiltration and inflammation in multiple organs, including lymph nodes, potentially influencing endothelial phenotype, adhesion molecular expression, and cytokine responsiveness [17,19,20,21,22]. These primary cells present opportunities for targeted drug screening, cytokine signaling studies, and modeling of endothelial barrier function under inflammatory conditions relevant to both normal equine physiology and MEED pathogenesis.

Long-term culturing of primary equine endothelial cells required optimized conditioning media to preserve cellular viability, proliferation, and phenotype. RPMI media supplemented with 10% FBS provided optimal conditions for sustained growth. The cells demonstrated stable proliferation for at least four passages without differentiation but did not survive beyond the thirteenth passage after trypsinization, establishing key parameters for standardized endothelial assays in equine research.

Our biomarkers analysis is summarized in Table 3. The cells were positive for CD31 (PECAM-1), ZO-1, and vimentin, which are known endothelial and mesenchymal markers, indicating intact intercellular junctions and cytoskeletal integrity [32,33,34]. E-cadherin and β-catenin positivities further support the presence of adherens junctions and suggest potential involvement in Wnt signaling pathways, relevant to both endothelial stability and immune interactions [33,35].

CD31 was selected as an endothelial marker with high specificity for vascular structures, while LYVE1 was incorporated to assess lymphatic endothelial identity. The co-expression of LYVE1 with CD31 points toward a lymphatic endothelial phenotype [36,37,38,39,40]. However, we acknowledge that definitive discrimination between lymphatic and blood vascular endothelium might require additional markers, such as PROXI or podoplanin (PDPN) [41,42]. Furthermore, given that the source tissue was obtained from a horse diagnosed with MEED, it is possible that disease-related immune activation might influence biomarker expression patterns. Experimental and clinical studies have shown that inflammatory cytokines and chronic immune activation can downregulate PROX1 and PDPN [43,44,45].

Interestingly, CD-79 positivity, which is typically associated with B-cell lineage, may reflect endothelial lymphocyte interactions within lymph nodes or specialized high endothelial venules (HEVs), which are vascular structures that help with lymphocyte trafficking [46,47,48,49,50,51]. Considering the disease context from which the tissue was derived, where chronic immune activation and eosinophilic infiltration are present, this could have driven atypical endothelial gene expression. This shift may reflect an adaptive response to enhance immune cell recruitment or pathological change. Therefore, CD79 in endothelial cells from lymph nodes warrants further study for its role in vascular remodeling and inflammation in horses. Lastly, negative stains for CD3, IBA1, and LY6G support the exclusion of T lymphocytes, macrophages, and neutrophils. Whole-genome sequencing of these primary cells achieved an average coverage of 1.87× with 98.88% alignment to the EquCab reference genome, confirming an equine origin and ruling out contamination. Although the coverage is below the 30× standard for comprehensive variant discovery, stringent filtering (>30 read depth, quality score of >20) yielded 107,385 high-confidence variants. These include 4891 missense variants and 45 stop-gained and 13 stop-lost mutations, all of which may have functional consequences.

Of particular note, numerous variants were found in USP25, a deubiquitinating enzyme involved in NF-κB signaling, immune regulation, and inflammatory response control [52]. Among these, an exonic split mutation was identified after filtration of read depths and quality score, the mutation was present at acritical exon–intron junction. This mutation could severely disrupt normal mRNA splicing, leading to frameshifting, and can lead to dysfunctional proteins. Given the central role that USP25 plays in regulating immune pathways, this variant could be mechanistically relevant to MEED pathogenesis, where dysregulated inflammation and immune activation are the hallmark features of this disease. It is, however, important to note that this genomic analysis represents primarily exploratory sequencing. While these results provide valuable insight into potentially relevant gene variants, follow-up validation using targeted Sanger sequencing and qPCR will be necessary to confirm variants’ presence and assess their expression or functional impact. Such validation will help establish a more direct link between genetic variation, endothelial phenotype, and disease processes like MEED.

This study establishes a reproducible protocol for isolating and characterizing primary equine lymph node-derived endothelial cells while also validating biomarkers and performing culture optimization, with supported genomic analysis. Biomarker profiling confirmed the endothelial identity with a likely lymphatic phenotype, and genomic sequencing verified the equine origin while identifying immune-related variants of potential relevance to MEED pathogenesis. The untransformed nature and disease-specific origin of these cells make them a valuable in vitro model for equine immunology and vascular biology. This work fills a critical gap in primary equine cell models by establishing a platform that combines molecular characterization, functional analysis, and disease relevance, enabling a targeted investigation into endothelial immune interactions in rare conditions such as MEED.

## Figures and Tables

**Figure 1 vetsci-12-00905-f001:**
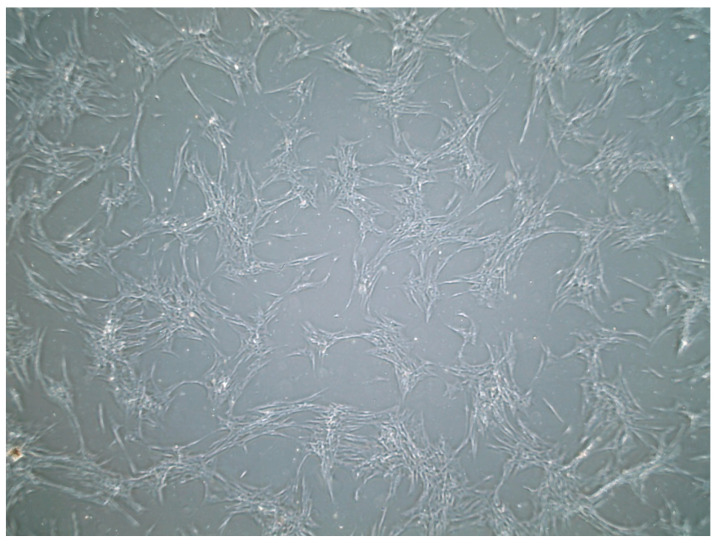
Equine cells at 20× magnification using an inverted microscope. These cells showed spindle-shaped morphology at passage 3, reaching 80% confluency within 5 days. No indications of contamination based on visual indicators with an inverted microscope were observed.

**Figure 2 vetsci-12-00905-f002:**
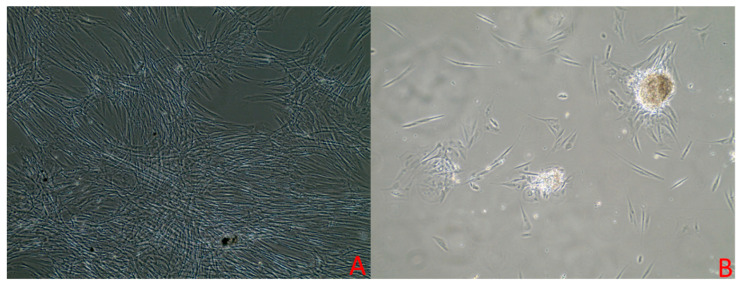
Equine cells at 20× magnification using an inverted microscope. Panel (**A**) shows cells at passage 8 at 85% confluency, exhibiting spindle-shaped morphology. Panel (**B**) shows cells at passage 5 at 60% confluency, demonstrating an increased number of epithelioid-shaped cells.

**Figure 3 vetsci-12-00905-f003:**
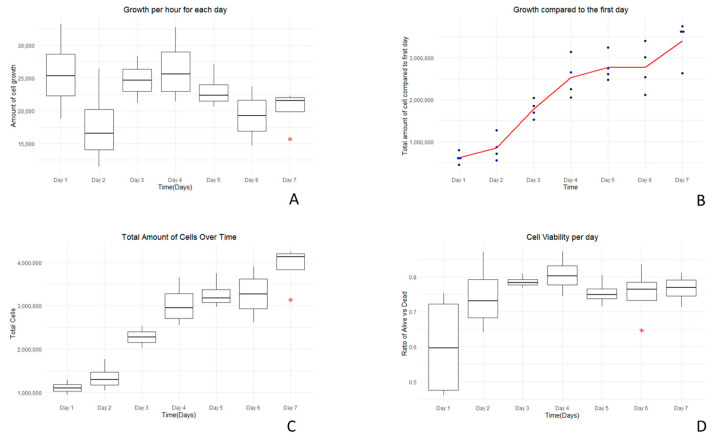
Cell growth dynamics. Cell viability was assessed using trypan blue. Graph (**A**) shows the growth rate per hour each day for 7 days. Graph (**B**) illustrates growth over time relative to the first day. Graph (**C**) presents the total number of cells over 7 days. Graph (**D**) illustrates the cell viability over time. Blue dots in graph B represent individual data values, while red asterisks denote potential outliers.

**Figure 4 vetsci-12-00905-f004:**
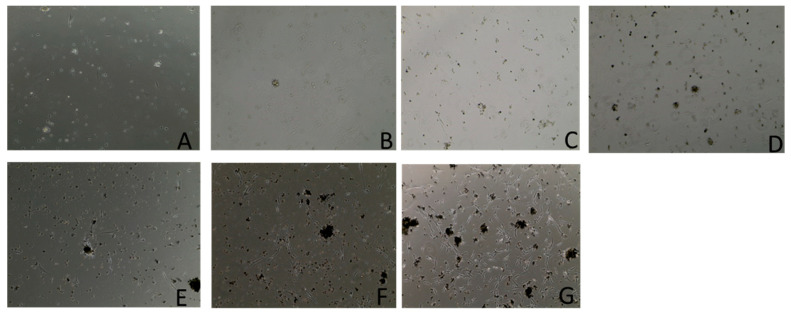
Cell images at 20× magnification: (**A**) 24 h after initial seeding; (**B**) 48 h after initial seeding; (**C**) 72 h after initial seeding; (**D**) 96 h after initial seeding; (**E**) 120 h after the initial seeding; (**F**) 144 h after initial seeding; (**G**) 168 h after initial seeding.

**Figure 5 vetsci-12-00905-f005:**
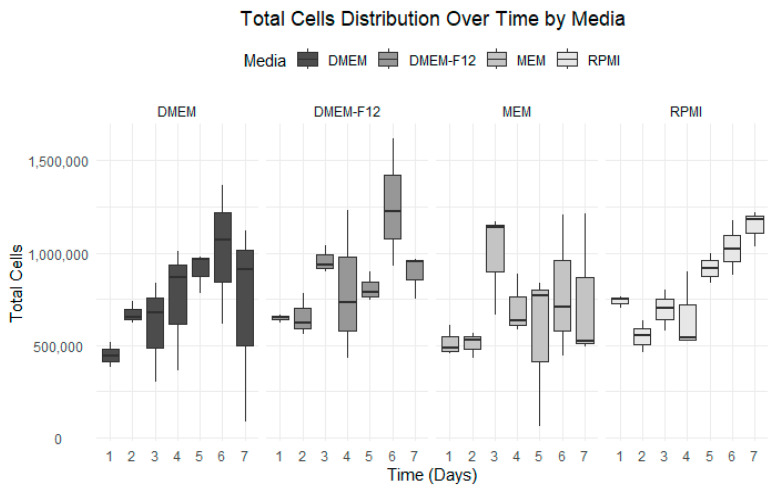
Media selection for cell growth optimization. Cell growth was monitored daily for 7 days, with each medium plotted separately to show growth trends.

**Figure 6 vetsci-12-00905-f006:**
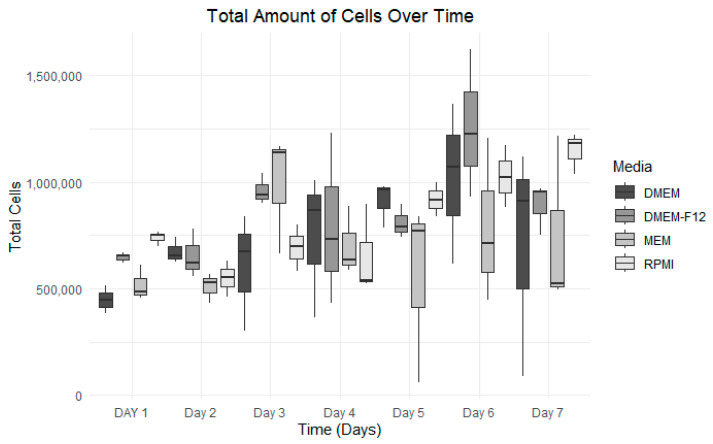
Total cell counts over time for each medium. Grayscale bars indicate the following: DMEM (dark gray), DMEM/F-12 (medium gray), MEM (light gray), and RPMI (lightest gray). Data from all media were plotted, showing a clear difference in cell count.

**Figure 7 vetsci-12-00905-f007:**
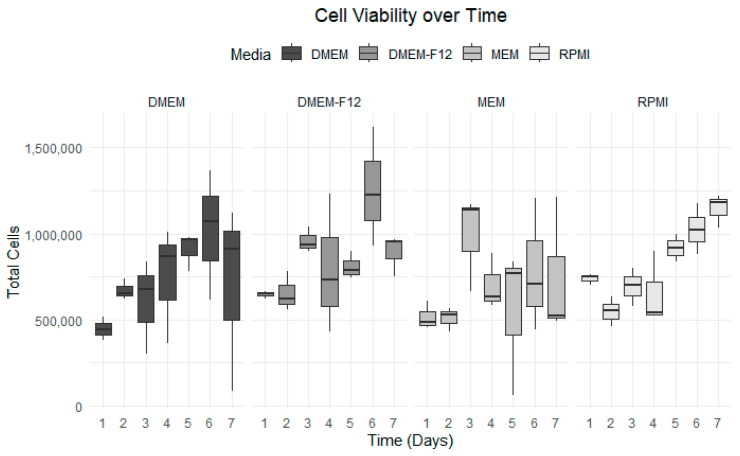
Cell viability over 7 days for each type of media. Grayscale bars indicate the following: DMEM (dark gray), DMEM/F-12 (medium gray), MEM (light gray), and RPMI (lightest gray). Viability was calculated daily as the ratio of live to dead cells.

**Figure 8 vetsci-12-00905-f008:**
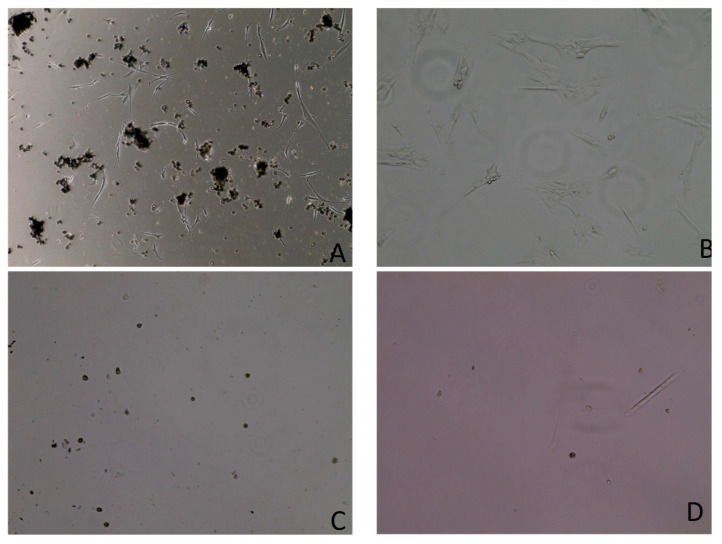
Primary cell culture in different types of media was imaged using an inverted microscope at 20× and 40×. (**A**) Day 7 post-seeding in RPMI medium at 20×. (**B**) Day 7 in DMEM/F-12 at 40×. (**C**) Day 7 in DMEM at 40×. (**D**) Day 7 in MEM at 40×.

**Figure 9 vetsci-12-00905-f009:**
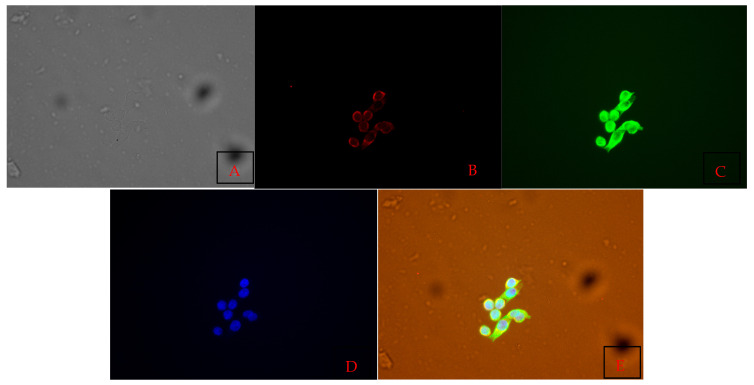
Primary cells were stained and imaged using a Plan Fluor 40× DIC M Lens. Panel (**A**) presents the brightfield morphology of the cultured cells. Panel (**B**) shows β-catenin localization (red), while Panel (**C**) depicts vimentin expression (green). Panel (**D**) visualizes nuclear DNA with DAPI staining (blue). Panel (**E**) presents the merged image, illustrating the combined distribution of β-catenin, vimentin, and DAPI within the same field of view.

**Figure 10 vetsci-12-00905-f010:**
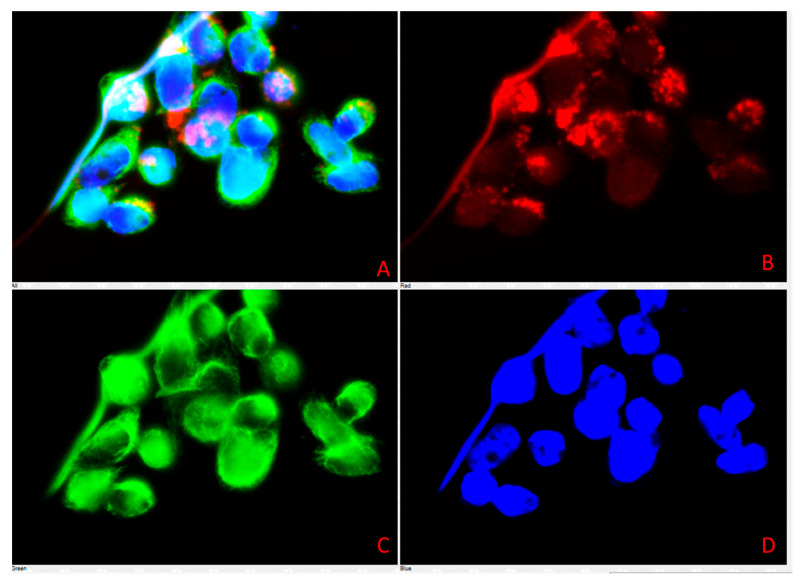
Primary cells were stained and imaged using Plan Fluor 40× DIC microscopy. Panel (**A**) is the merged image of all 3 fluorescence signals. Panel (**B**) shows CD31 in red, Panel (**C**) shows vimentin in green, and Panel (**D**) shows DAPI in blue.

**Figure 11 vetsci-12-00905-f011:**
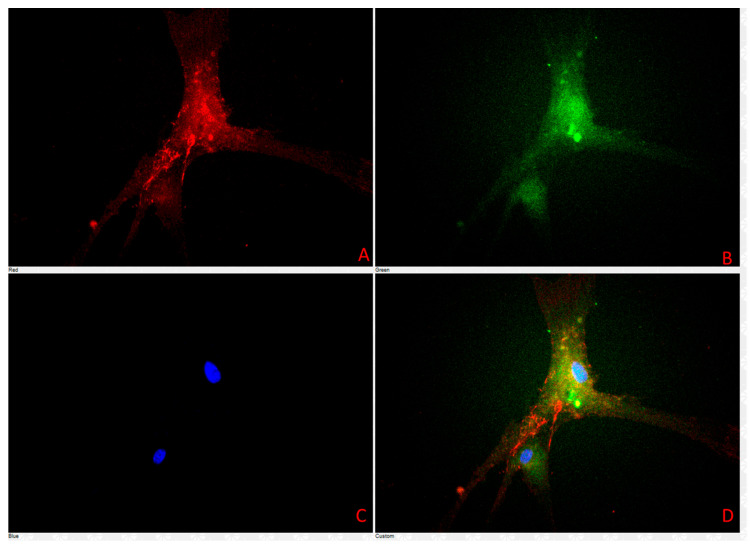
Primary cells were stained and imaged using Plan Fluor 40× DIC M microscopy. Panel (**A**) shows ZO-1 in red, Panel (**B**) shows vimentin in green, Panel (**C**) shows DAPI in blue, and Panel (**D**) shows all 3 fluorescence signals.

**Figure 12 vetsci-12-00905-f012:**
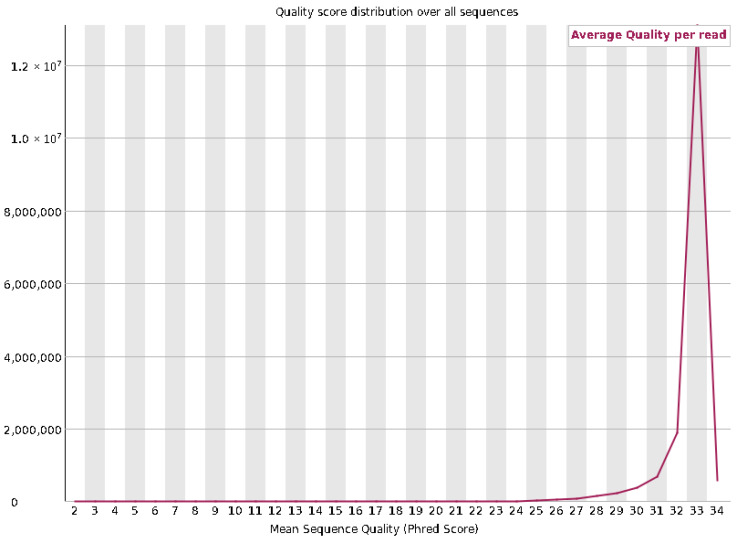
Quality score distribution across all sequences. The plot shows the mean sequence quality (Phred score) distribution, with the majority of reads having an average quality score around Q33.

**Figure 13 vetsci-12-00905-f013:**
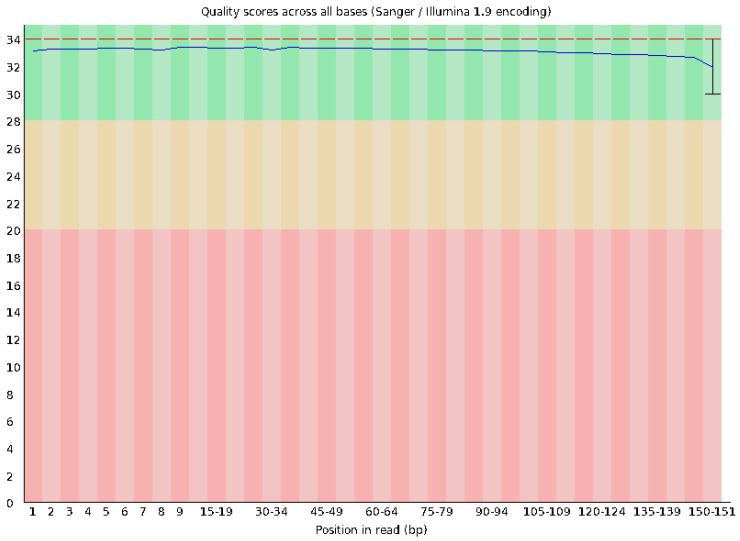
Quality scores across all bases. The plot illustrates the base quality scores for each position in the read, with the majority of bases maintaining a high Phred score above Q30 throughout the length of the read. Red line indicates the average quality of reads at each position while the blue line is the median quality at each position.

**Table 1 vetsci-12-00905-t001:** Growth of primary cells over 7 days. Cell viability was assessed using trypan blue assay. Each value represents the mean from three independent replicates (*n* = 3) for each parameter: average cells, average dead cells, average live cells, and average of total cells.

Growth Curve Assay in RPMI Media				
Number of Days	Average Amount of Cells	Average Amount of Dead Cells	Average Amount of Live Cells	Average Amount of Total Cells
Day 1	1.12 × 10^6^	5.56 × 10^5^	8.39 × 10^5^	1.40 × 10^6^
Day 2	1.35 × 10^6^	4.86 × 10^5^	1.20 × 10^6^	1.69 × 10^6^
Day 3	2.28 × 10^6^	4.44 × 10^5^	1.63 × 10^6^	2.07 × 10^6^
Day 4	3.03 × 10^6^	5.04 × 10^5^	2.25 × 10^6^	2.75 × 10^6^
Day 5	3.27 × 10^6^	5.78 × 10^5^	1.76 × 10^6^	2.34 × 10^6^
Day 6	3.27 × 10^6^	5.55 × 10^5^	1.78 × 10^6^	2.34 × 10^6^
Day 7	3.91 × 10^6^	6.49 × 10^5^	2.14 × 10^6^	2.80 × 10^6^

**Table 2 vetsci-12-00905-t002:** Growth kinetics of primary cell culture derived from equine lymph node in four different media over a 7-day period.

Number of Days	DMEM	DMEM-F12	MEM	RPMI
Day 1	4.48 × 10^5^	6.48 × 10^5^	5.18 × 10^5^	7.39 × 10^5^
Day 2	6.73 × 10^5^	6.55 × 10^5^	5.10 × 10^5^	5.49 × 10^5^
Day 3	6.04 × 10^5^	9.59 × 10^5^	9.91 × 10^5^	6.93 × 10^5^
Day 4	7.47 × 10^5^	7.97 × 10^5^	7.05 × 10^5^	6.53 × 10^5^
Day 5	9.10 × 10^5^	8.10 × 10^5^	5.56 × 10^5^	9.17 × 10^5^
Day 6	1.02 × 10^6^	1.26 × 10^6^	7.88 × 10^5^	1.03 × 10^6^
Day 7	7.07 × 10^5^	8.91 × 10^5^	7.44 × 10^5^	1.15 × 10^6^

**Table 3 vetsci-12-00905-t003:** Summary of biomarker detection using immunofluorescence assay.

Biomarker	Positive	Negative
CD-3		Negative
CD-31	Positive	
CD-79	Positive	
E-cadherin	Positive	
Vimentin	Positive	
Beta-Catenin	Positive	
IBA1		Negative
LY6G		Negative
LYE1	Positive	

**Table 4 vetsci-12-00905-t004:** Overview of variant counts by category. This table shows the total number of biallelic and multiallelic SNVs, indels, and STRs, along with their respective genotype calls and coding variants.

Category	Count	Percentage
** Total reads**	16,855,223	100.00%
** Paired reads**	16,855,223	100.00%
**Concordant alignments**		
Aligned concordantly 0 times	3,758,073	22.30%
Aligned concordantly exactly 1 time	10,345,915	61.38%
Aligned concordantly >1 time	2,751,235	16.32%
**Discordant alignments**		
Pairs aligned concordantly 0 times	3,758,073	
Aligned discordantly 1 time	2,748,824	73.14%
**Unaligned pairs**		
Pairs aligned neither concordantly nor discordantly	1,009,249	
Mates making up these pairs	2,018,498	
Aligned 0 times	379,206	18.79%
Aligned exactly 1 time	323,314	16.02%
Aligned >1 time	1,315,978	65.20%
Overall alignment rate	98.88%	

**Table 5 vetsci-12-00905-t005:** Alignment summary of paired reads. A total of 16,855,223 paired reads were processed, with an overall alignment rate of 98.88%. Of these, 22.30% of paired reads did not align concordantly, while 61.38% aligned concordantly exactly once and 16.32% aligned concordantly more than once. Among the pairs that failed to align concordantly, 73.14% aligned discordantly at least once. For the remaining unaligned pairs, 18.79% of reads did not align at all, 16.02% aligned exactly once, and 65.20% aligned more than once.

Category	Biallelic SNVs	Biallelic Indels	Biallelic STRs	Other Biallelic	Multiallelic SNVs	Multiallelic Indels	Multiallelic STRs	Other Multiallelic
Variants	1,202,674	264,537	1366	0	220	692	942	0
Genotype calls	4859	316	23	0	25	14	10	0
Coding variants	107,385	23,754	123	0	15	55	93	0

**Table 6 vetsci-12-00905-t006:** This table presents the results of the biallelic SNV analysis, detailing the total variants identified, including coding variants, synonymous and missense mutations, and significant mutations, such as stop-lost, stop-gained, and start-lost. Splice site mutations, along with exonic and splice region variants, are also reported. Non-coding variants are included for the 5′ and 3′ UTRs and intronic and intergenic regions.

Category	Count
Total Biallelic SNVs	1,202,674
Coding Variants	107,385
Synonymous Variants	5499
Missense Variants	4891
Stop-Lost Variants	13
Stop-Gained Variants	45
Start-Lost Variants	3
Splice Donor Variants	24
Splice Acceptor Variants	17
Exonic Splice Region Variants	154
Splice Region Variants	1062
5′ UTR Variants	80,899
3′ UTR Variants	14,778
Upstream Transcript Variants	10,368
Downstream Transcript Variants	3117
Intron Variants	357,493
Intergenic Variants	724,311

## Data Availability

The raw sequencing data generated in this study have been deposited in the NCBI Sequence Read Archive (SRA) under accession number PRJNA1295672 for BioProject. These data are publicly available and can be accessed through the SRA database at https://dataview.ncbi.nlm.nih.gov/object/SRR34697451, accessed on 9 September 2025.

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
