# Peer review of "Isolation and Characterization of Equine Lymph Node Endothelial Cells"

_vetsci, 2025, doi:10.3390/vetsci12090905_

Round 1
Reviewer 1 Report
Comments and Suggestions for Authors
Introduction notes:
The authors introduce the paper describing basic background of lymph nodes as well as in vitro models. The authors declare that they have established an untransformed novel equine lymph node primary cell but conclude the introduction discussing MEED. This is confusing and suggests to the reader that this manuscript is about MEED and not the development of a cell line that could help enhance the research of the molecular mechanisms underlying MEED. I recommend the authors include a closing paragraph in the introduction that concisely states why there is a need for the cell line as well as describing the characterization efforts (growth curves, biomarkers, sequencing, etc). The paragraph should be end with a statement similar to the last statement in the abstract (lines 17-19).
Methodology notes:
2.1 includes some description of results which should be removed from this section (lines 70-80).
Line 111 – “or passing” should read “or passaging”
Results notes:
Table 1: States experiments were performed in triplicate but table shows N=4
Discussion notes:
References are needed for lines 360-366
Line 368 “thatthat” delete one that
Line 384 “interesting” should be interestingly, xyz
Needs a conclusion paragraph summarizing the research approach, data analysis, and results gleaned that indicate why/how this primary cell line protocol is appropriate for in vitro assays.
Also, no other mention of MEED except for introduction. This would have an impact on research applicability and potential phenotypes observed so it should be discussed.
Comments on the Quality of English LanguageNeeds improvement in English: Lines 16, 27-28, 33, 35, 40, 57-58, 59, 112 (stopped keeping count, please address English throughout)
Author Response
Comment 1: Lines 50–62: The text discusses Multisystemic Eosinophilic Epitheliotropic Disease (MEED), which is said to affect the animal “donor” of the studied cells. Considering that the objective of the study is the description of the cells obtained, this disease description is not relevant to the manuscript, unless the authors provide a justification (i.e., a connection between the studied cells and MEED).
Response 1: Thank you for your comments and feedback. We have revised the introduction to clearly explain the connection between MEED and the primary cells described in this study. MEED involves eosinophilic infiltration and inflammation within lymphoid tissues, the lymph node derived endothelial cells obtained from this case provide a disease-relevant and physiologically meaningful model for both immunological and pathogenesis research. To make this connection explicit, we have added content in lines 99–107 that highlights the study’s aim and emphasizes the relevance of these cell lines as a steppingstone for MEED research and other equine immunological investigations
Methodology notes:
Comment 2: 2.1 includes some description of results which should be removed from this section (lines 70-80).
Response 2: Sections were removed Line133-15 “Both sections exhibit marked dermal thickening with focal epidermal ulceration and deep eosinophilic infiltration extending into the subcutis. On cut section, the dermis appears firm and white to tan, while the subcutis is thickened and mottled tan to dark red”
Comment 3: Line 111 – “or passing” should read “or passaging”
Response 3: Corrected passaging
Results notes:
Comment 4: Table 1: States experiments were performed in triplicate but table shows N=4
Response 4: Thank you for that observation. The “N=4” notation was a misunderstanding caused by the table format. The four columns represent separate parameters average total cells, average dead cells, average live cells, and average of total cells rather than four replicates. All values are the mean of three independent replicates per day. To clarify, we have updated the table legend to explicitly state the number of replicates. “Revised table: The growth of the primary cells over 7 days. Cell viability was assessed using trypan blue assay. Each value represents the mean from three independent replicates (N=3) for each parameter: average total cells, average dead cells, average live cells, and average of total cells.”
Discussion notes:
Comment 5: References are needed for lines 360-366
Response 5: We have added appropriate citations to support the statements in these lines and rewrote this section to add clear detail and supportive arguments.
Comment 6: Line 368 “thatthat” delete one that
Line 384 “interesting” should be interestingly,
Response 6: We have corrected “thatthat” to “that” Line 368 and changed “interesting” to “interestingly”
Comment 7: Needs a conclusion paragraph summarizing the research approach, data analysis, and results gleaned that indicate why/how this primary cell line protocol is appropriate for in vitro assays.
Response 7: We have added a conclusion summarizing the research approach, biomarker analysis, genomic findings, and culture optimization results. This new paragraph highlights how and why the protocol is suitable for in vitro assays, with specific reference to its reproducibility, phenotypic stability, and equine relevance
Comment 8: Also, no other mention of MEED except for introduction. This would have an impact on research applicability and potential phenotypes observed so it should be discussed.
Response 8: We have now incorporated MEED-related discussion into the biomarker interpretation section (including the potential disease-related modulation of markers such as PROX1 and PDPN) and in the genomic analysis section, emphasizing the possible relevance of USP25 variants to MEED pathogenesis.
Reviewer 2 Report
Comments and Suggestions for Authors
Dear authors,
Very interesting and well presented manuscript. Horses do present also changes in the small intestine with this disease, some times these lesions can be surgical.
On figure 11, you state "equine primary cells" but you most write "lymph-node endothelial cells"
KR
Author Response
Comment 1: Very interesting and well presented manuscript. Horses do present also changes in the small intestine with this disease, some times these lesions can be surgical.
Response 1: Thank you for your comments and yes horses do sometimes present changes in small intestine with MEED Disease therefore, we decided to add line 90-95 “In addition to these clinical signs, reports in certain horse breeds have documented lesions in the intestines, stomach, and cecum. Histopathological examination of surgically excised tissue revealed lymphoplasmacytic infiltration, and immunohistochemistry demonstrated CD3-positive cell populations. 1,2,3 In contrast, three reported cases in donkeys did not describe gastric lesions; instead, lesions were observed in the colon and colonic lymph nodes. 3
Comment 2: On figure 11, you state "equine primary cells" but you most write "lymph-node endothelial cells"
Response 2: Reword the figure statement to include “Equine primary lymph-node cells
Citations:
(1) Horan, E. M., Metcalfe, L. V. A., de Swarte, M., Cahalan, S. D. & Katz, L. M. Pulmonary and hepatic eosinophilic granulomas and epistaxis in a horse suggestive of multi-systemic eosinophilic epitheliotropic disease. Equine Vet Educ 25, (2013).
(2) Bosseler, L. et al. New Zealand Veterinary Journal Equine multisystemic eosinophilic epitheliotropic disease: A case report and review of literature Equine multisystemic eosinophilic epitheliotropic disease: A case report and review of literature. (2013) doi:10.1080/00480169.2012.753569.
(3) Paraschou, G., Vogel, P. E., Lee, A. M., Trawford, R. F. & Priestnall, S. L. Multisystemic eosinophilic epitheliotropic disease in three donkeys. J Comp Pathol 201, (2023).
Reviewer 3 Report
Comments and Suggestions for Authors
This work is very interesting because it isolates equine cells in vitro. This offers the
possibility of conducting experiments with a model closer to real-life conditions than cell
lines. The experimental design is well-designed, although I believe that some additional markers, such as
von Willebrand factor, should have been added. The results are presented clearly, using
appropriate graphs and figures.
However, some aspects need improvement.
The introduction presents lymph nodes in a very strange way, dividing them into layers.
It is a basic concept in histology that solid organs are divided into parenchyma and stroma, and into regions; layers are used for hollow organs (with the exception of the organs of the oral and nasal cavities). Even in the bibliography the authors use for this statement, layers are not mentioned in the description of lymph nodes.
The discussion requires further investigation; the literature used is very limited and,
in some cases, very general. It would be important to discuss in greater detail whether the
endothelial population could be lymphatic or blood vascular endothelium. The authors should
explain why they did not use certain markers that could allow them to differentiate between the
two cell types.
Author Response
Comment 1: This work is very interesting because it isolates equine cells in vitro. This offers the
possibility of conducting experiments with a model closer to real-life conditions than cell
lines.
Response 1: Thank you for your comments and feedback.
Comment 2: The experimental design is well-designed, although I believe that some additional markers, such as von Willebrand factor, should have been added.
Response 2: Yes, von Willebrand factor could be useful in further confirming the presence of endothelial cell markers; however, its expression can be significantly influenced by various diseases, particularly bleeding disorders, where studies have shown notable alterations in expression levels.1,2 Although MEED disease has shown no signs of hemorrhage to date, there is limited information available on this condition. Notably, one case study described blood discharge from the nostril, accompanied by blood test results indicating thrombocytopenia, but with no evidence of any other bleeding disorder.3 Given the scarcity of data on MEED disease, we elected to focus on more ubiquitous endothelial markers, such as CD31 and LYVE1.
Comment 3: The results are presented clearly, using appropriate graphs and figures. However, some aspects need improvement.
Response 3: Thank you for your comments. While it was not entirely clear which aspects of the results required improvement, we revised the figure captions to enhance clarity and also refined the results section to present the information more clearly.
Comment 4: The introduction presents lymph nodes in a very strange way, dividing them into layers. It is a basic concept in histology that solid organs are divided into parenchyma and stroma, and into regions; layers are used for hollow organs (with the exception of the organs of the oral and nasal cavities). Even in the bibliography the authors use for this statement, layers are not mentioned in the description of lymph nodes.
Response 4: We agree with your comments and have revised the introduction to accurately reflect the correct histological terminology for lymph nodes. Previous section was replaced with the following paragraph : Line 42- 47 “Lymph nodes are complex structures involved in the immune system of most mammalian species.4–6 They are made up of parenchyma containing lymphocytes and other immune cells, such as natural killer cells and macrophages. 7 This cellular component is supported by a stromal framework that provides structural integrity. The parenchyma is organized into distinct regions which are the cortex containing lymphoid follicles, paracortex and medulla.8 Lymph nodes function as filters for lymphatic fluid for monitoring pathogens and serve as sites for the activation and proliferation of immune cells.4,7.”
Comment 5: The discussion requires further investigation; the literature used is very limited and,
in some cases, very general.
Response 5: We agree that further investigation and discussion are warranted. We have added two paragraphs: the first discusses the usefulness of these primary cells and their applications in advancing the understanding of MEED disease, and the second explains other potential biomarkers and the rationale for selecting the ones used in our study. In addition, we have included a paragraph addressing whether the endothelial population could be lymphatic or blood vascular in origin.
Comment 6: It would be important to discuss in greater detail whether the endothelial population could be lymphatic or blood vascular endothelium. The authors should explain why they did not use certain markers that could allow them to differentiate between the two cell types.
Response 6: We agree that it is important to address the source of the endothelial population. We have added a section and paragraph discussing the use of CD31 and LYVE1, along with other biomarkers such as cadherin and their application. We also included discussion on how certain biomarkers may vary due to the nature of the animal’s disease.
Reviewer 4 Report
Comments and Suggestions for Authors
In this manuscript, the authors report the isolation and characterization of endothelial cells from an equine lymph node. After a careful reading of the manuscript, we observed the following points:
Lines 50–62: The text discusses Multisystemic Eosinophilic Epitheliotropic Disease (MEED), which is said to affect the animal “donor” of the studied cells. Considering that the objective of the study is the description of the cells obtained, this disease description is not relevant to the manuscript, unless the authors provide a justification (i.e., a connection between the studied cells and MEED). It is suggested that, in the Introduction, the authors describe the evaluation/characterization criteria for the cells in question, and at the end of the Introduction, clearly state the aim of the investigation.
Lines 69–80 (Methods): Results are described in this section, which not only belong to the wrong section but also seem to relate to the clinical condition of the donor animal rather than to the study’s objective. These should be removed.
Lines 120–121: A starting point of 500,000 is mentioned; however, in line 207, the initial seed is stated to be 50,000 cells.
Line 187: This sentence belongs in item 3.1.
Lines 190–191: It is stated that Figure 1 shows cells with an epithelioid shape; however, the figure legend states that “these cells showed spindle-shaped morphology.”
Line 206: Where is Graph 1?
Lines 213–215: This text belongs in the Materials and Methods section. The legend should be rewritten.
Table 1: The table is not understandable; column titles are likely missing.
Lines 217–220 and 222–224: Rewrite these legends to avoid excessive repetition of words.
Standardize the presentation of results: In item 3.2, the table is presented first, followed by small graphs, and finally the figures. In item 3.3, the graphs are presented before the table, and separately.
Item 3.4 (Cell Characterization Using Immunofluorescence Assay): There is simply no description of the results. This section must be rewritten.
Item 3.5: The presentation of the results is poor. The tables are not mentioned in the text.
Discussion: The discussion must be rewritten, as it repeats sections from the Introduction and Results.
Finally, several references are irrelevant to the study, such as references 11 to 15.
In our view, extensive revision is required before this manuscript can be considered for publication.
Author Response
Comment 1: In this manuscript, the authors report the isolation and characterization of endothelial cells from an equine lymph node. After a careful reading of the manuscript, we observed the following points: Lines 50–62: The text discusses Multisystemic Eosinophilic Epitheliotropic Disease (MEED), which is said to affect the animal “donor” of the studied cells. Considering that the objective of the study is the description of the cells obtained, this disease description is not relevant to the manuscript, unless the authors provide a justification (i.e., a connection between the studied cells and MEED). It is suggested that, in the Introduction, the authors describe the evaluation/characterization criteria for the cells in question, and at the end of the Introduction, clearly state the aim of the investigation.
Response 1: Thank you for your comments and feedback. We have revised the introduction to include relevant information specific to this study and clarified the connection between MEED and the cell line (lines 79–87). We have also added a clear statement of the study’s aim (lines 85–87) and specified the criteria used for cell characterization (lines 84–85). We revised the language and structure of the introduction to improve clarity, wording, and overall flow.
Comment 2: Lines 69–80 (Methods): Results are described in this section, which not only belong to the wrong section but also seem to relate to the clinical condition of the donor animal rather than to the study’s objective. These should be removed.
Response 2: We agree with the reviewer the sections were removed lines 69-80
Comment 3: Lines 120–121: A starting point of 500,000 is mentioned; however, in line 207, the initial seed is stated to be 50,000 cells.
Response 3: Fix typo to 500,000 cells
Comment 4: Line 187: This sentence belongs in item 3.1.
Response 4: Moved this sentence to 3.1
Comment 5: Lines 190–191: It is stated that Figure 1 shows cells with an epithelioid shape; however, the figure legend states that “these cells showed spindle-shaped morphology.”
Response 5: Figure 1 was corrected to figure 2.B
Comment 6: Line 206: Where is Graph 1?
Response 6: Graph 1 was changed to Figure 3. This is corrected in the paper.
Comment 7: Lines 213–215: This text belongs in the Materials and Methods section. The legend should be rewritten.
Response 7: Lines 213–215: This text belongs in the Materials and Methods section. The legend should be rewritten.
Comment 8: Table 1: The table is not understandable; column titles are likely missing.
Response 8: Rewrote the legend to “Growth of primary cells over 7 days. Cell viability was assessed using trypan blue assay. Each value represents the mean from three independent replicates (N=3) for each parameter: average total cells, average dead cells, average live cells, and average of total cells.” The column titles are Number of Days / Average Amount of Cells Average/ Amount of Dead Cells /Average Amount of Live Cells/ Average Amount of Total Cells. Please let us know if the titles are not populated on your view.
Comment 9: Lines 217–220 and 222–224: Rewrite these legends to avoid excessive repetition of words.
Response 9: Rewrote the legends
Comment 10: Standardize the presentation of results: In item 3.2, the table is presented first, followed by small graphs, and finally the figures. In item 3.3, the graphs are presented before the table, and separately.
Response 10: Change order of the graphs to standardize the results.
Comment 11: Item 3.4 (Cell Characterization Using Immunofluorescence Assay): There is simply no description of the results. This section must be rewritten.
Response 11: This section was rewritten.
Comment 12: Item 3.5: The presentation of the results is poor. The tables are not mentioned in the text.
Response 12: The tables were added and mentioned in the text.
Comment 13: Discussion: The discussion must be rewritten, as it repeats sections from the Introduction and Results. Finally, several references are irrelevant to the study, such as references 11 to 15.
In our view, extensive revision is required before this manuscript can be considered for publication.
We have extensively revised the discussion to eliminate redundancy with the introduction and results; it now focuses on interpreting the data through the aim of the study objectives and relevant literature. For example, sections that repeated the biomarker descriptions from the results have been condensed into a brief summary before moving into their functional interpretation. Similarly, genomic variant listings were shortened and reframed to emphasize their potential biological relevance rather than representing raw data. The structure has been reorganized to open with the research gap, followed by a concise overview of our approach and main findings, an interpretation framed within existing literature, an expanded discussion of MEED-related implications, and a concluding section on the novelty and applicability of this endothelial model for equine in-vitro assays.
Round 2
Reviewer 1 Report
Comments and Suggestions for Authors
VetSci-3808063- peer review revision 2 notes
Excellent revised manuscript, just minor grammatical suggestions for consideration by the authors.
Line 77: Replace “Its” with “Overall, MEED”
Line 79: Remove “a” after establishing
Line 80: add “horses afflicted by” in front of MEED
Line 197: add “was” in front of subsequently
Line 197-198: Replace the period following trypsinizatin with a comma and remove “Thus, the process”. Also replace “led” with “leading” and “originated” with “originating”
Line 203: Add parentheses around Figure 1
Line 204: Replace “Once” with “Upon”
Line 209: Add “as” in front of shown
Line 212: Add comma following passage 3 and replace “in” to “within”
Line 213: add “were observed” following inverted microscope.
Line 219: Change to “The results of the growth curve assays are…”
Figure 4 images: Authors may want to consider lightening the background of the images in A, D, E, F, and G, as they are still very dark which makes visualizing the cells difficult.
Figure 8 images: same comment as in Fig 4.
Figure 9 A: Authors may want to consider adding more contrast/lightening the background to allow the reader to more easily visualize the cells.
Line 365: Remove “The”, capitalize “Untransformed” , change “address the” to “addresses this”
Line 366: add “knowledge by” in front of providing
Line 367: Change “investing” to “investigating”
Line 368: remove “in”
Line 375: add “cellular” in front of viability
Line 416: Replace “frameshift” with “frameshifting”
Line 417: Change to “Given the central role that USP25 plays in regulating…”
Line 418: Change “pathway” to “pathways,”
Line 435: Change “interaction” to “interactions”
Author Response
Excellent revised manuscript, just minor grammatical suggestions for consideration by the authors.
Line 77: Replace “Its” with “Overall, MEED”
Replaced with “Overall MEED”
Line 79: Remove “a” after establishing
Removed a
Line 80: add “horses afflicted by” in front of MEED
Added “Horses afflicted by”
Line 197: add “was” in front of subsequently
Add “was”
Line 197-198: Replace the period following trypsinizatin with a comma and remove “Thus, the process”. Also replace “led” with “leading” and “originated” with “originating”
Deleted thus the process and replace led to leading and originated to originating
Line 203: Add parentheses around Figure 1
Added parentheses
Line 204: Replace “Once” with “Upon”
Reworded to Upon
Line 209: Add “as” in front of shown
Add “as”
Line 212: Add comma following passage 3 and replace “in” to “within”
Add a comma and replace to within
Line 213: add “were observed” following inverted microscope.
Add “were observed”
Line 219: Change to “The results of the growth curve assays are…”
Changed to line above
Figure 4 images: Authors may want to consider lightening the background of the images in A, D, E, F, and G, as they are still very dark which makes visualizing the cells difficult.
Figure 8 images: same comment as in Fig 4.
Increase the brightness in the TIFF file
Figure 9 A: Authors may want to consider adding more contrast/lightening the background to allow the reader to more easily visualize the cells.
Line 365: Remove “The”, capitalize “Untransformed” , change “address the” to “addresses this”
Changed to “Untransformed primary culture, derived from lymph nodes, addresses the gap in knowledge…”
Line 366: add “knowledge by” in front of providing
Added knowledge
Line 367: Change “investing” to “investigating”
changed
Line 368: remove “in”
removed
Line 375: add “cellular” in front of viability
Added Cellular
Line 416: Replace “frameshift” with “frameshifting”
Changed to frameshifting
Line 417: Change to “Given the central role that USP25 plays in regulating…”
Changed
Line 418: Change “pathway” to “pathways,”
Add a s in pathways
Line 435: Change “interaction” to “interactions”
Changed to interactions
Reviewer 3 Report
Comments and Suggestions for Authors
In the present form the manuscript can b accepted
Author Response
In the present form the manuscript can be accepted:
Thank you for your comment.